# Operando Laboratory X-ray Absorption Spectroscopy and UV–Vis Study of Pt/TiO₂ Photocatalysts during Photodeposition and Hydrogen Evolution Reactions

Elizaveta G. Kozyr [1,2], Peter N. Njoroge [1], Sergei V. Chapek [1], Viktor V. Shapovalov [1], Alina A. Skorynina [1], Anna Yu. Pnevskaya [1], Alexey N. Bulgakov [1], Alexander V. Soldatov [1], Francesco Pellegrino [2], Elena Groppo [2], Silvia Bordiga [2], Lorenzo Mino [2] and Aram L. Bugaev [1,3,*]

1   The Smart Materials Research Institute, Southern Federal University, Sladkova 178/24, 344090 Rostov-on-Don, Russia
2   Department of Chemistry, University of Turin, Via Giuria 7, 10125 Torino, Italy
3   Paul Scherrer Institute, Forschungsstrasse 111, 5232 Villigen, Switzerland
*   Correspondence: abugaev@sfedu.ru

**Abstract:** Photocatalytic hydrogen ($H_2$) production is a promising route for alternative energetics. Understanding structure–activity relationships is a crucial step towards the rational design of photocatalysts, which requires the application of operando spectroscopy under relevant working conditions. We performed an operando investigation on a catalytic system during the photodeposition of Pt on TiO₂ and photostimulated $H_2$ production, using simultaneous laboratory X-ray absorption spectroscopy (XAS), UV–Vis spectroscopy, and mass spectrometry. XAS showed a progressive increase in Pt fluorescence for Pt deposited on TiO₂ for over an hour, which is correlated with the signal of the produced $H_2$. The final Pt/TiO₂ catalyst contained Pt(0) particles. The electronic features corresponding to the Pt$^{4+}$ species in the UV–Vis spectrum of the solution disappear as soon as UV radiation is applied in the presence of formic acid, which acts as a hole scavenger, resulting in the presence of Pt(0) particles in solution.

**Keywords:** operando spectroscopy; XANES; photocatalysis; platinum catalyst; UV–Vis; HER; photodeposition

## 1. Introduction

Hydrogen is an attractive alternative energy source due to the possibility of obtaining it through green and renewable processes [1]. Much research has been performed on the splitting of water for hydrogen production [2] and hydrogen evolution reaction (HER). This is a remarkable and promising technology due to its lack of $CO_2$ emission and air pollution [3]. Pt deposited on TiO₂ has been one of the most commonly reported photocatalysts since the pioneering works in the field of water photodecomposition were published [4,5]. This material continues to motivate intensive research aimed toward understanding the interaction between semiconductors and metals and the mechanisms of metal-nanoparticle formation and incorporation on TiO₂, also considering the effects of different TiO₂ morphologies [6–8].

Operando spectroscopic techniques provide efficient tools for studying catalysts in real time under working conditions [9]. In particular, X-ray absorption spectroscopy (XAS) selectively probes the local structure around a specific element, which can be used for active site monitoring and to reveal the structural properties of metal species of photocatalysts during synthesis and reaction conditions. The main challenges for XAS measurements in photocatalytic reactions are that they are typically carried out in the liquid phase (which absorbs X-ray photons) and under UV or visible light irradiation (which requires the

development of suitable cells, allowing for simultaneous irradiation with X-ray and UV–Vis beams). A number of photocatalytic cells overcoming these issues have been proposed recently, and measurements of photocatalytic systems in operando conditions are now feasible [10,11]. There are many operando XAS studies on Pt-based systems [12–20]; nevertheless, the examples of operando XAS experiments on Pt that is supported by $TiO_2$ are still rare. Khare et al. [21] reported on the technical possibilities of in situ XAS during photodeposition/photocatalysis. Piccolo et al. [22] studied Pt deposited on $TiO_2$ by XAS under UV–visible light irradiation and showed that the system is simple and optimal in terms of electronic and catalytic stability while maximizing the photocatalytic hydrogen evolution reaction (PHER) efficiency. Ying Zhou et al. [23] performed an operando XAS experiment to track the interplay of Pt and crystal facets of $TiO_2$ during the oxidation of CO. In their work [21], operando XAS was used to follow the oxidation state of Pt in $Pt/TiO_2$ during the $H_2$ evolution reaction.

In addition, UV–Vis spectroscopy is sensitive to the electronic transitions of valence electrons and can be sensitive to changes in the oxidation state and/or ligand surrounding metals. A number of works have been devoted to studying the process of hydrogen production with $TiO_2$ photocatalysts containing Pt using UV–Vis spectroscopy [24–26].

Combining multiple spectroscopic techniques in one experimental setup offers the possibility of observing catalytic systems from different perspectives that provide complementary information. Tinnemans et al. [27] revealed the possibilities of combining several operando techniques in one spectroscopic-reaction cell. Their work [28] was devoted to operando DRIFTS combined with HERFD–XANES and XES to study the mechanisms behind photothermal catalytic oxidation of CO over $Pt/TiO_2$. The methods turned out to be sensitive enough to uncover a change in the electronic structure of the Pt sites upon light illumination. In another work [29], the "design gap" was eliminated by using a reaction vessel, where the addition of reactants and their stirring and mixing were carried out, while XAS and UV–Vis spectra were recorded in a spatially separated measurement cell. Yoshida et al. [30] attempted an ex situ XAS and UV–Vis of $Pt/TiO_2$ at different times of photodeposition. Although the potential of combined operando UV–Vis and XAS under working conditions is huge [31–33], there are still only a few studies devoted to the mechanisms of the photodeposition process and the structure of photodeposited metal particles.

In the current work, we used a multi-technique approach to study catalyst formation and photocatalytic $H_2$ production over $Pt/TiO_2$ by combining laboratory X-ray absorption spectroscopy (XAS), UV–Vis spectroscopy, and mass spectrometry (MS). A self-designed, 3D-printed cell was used to perform operando laboratory characterization. From the XAS data, the oxidation state of the Pt in the formed catalyst was determined, while the absolute intensity of the X-ray fluorescence signal was used to monitor the amount of Pt incorporated in $TiO_2$. UV–Vis spectra were used to track the Pt species in the solution, while MS data showed the evolution of the produced $H_2$ signal.

## 2. Results and Discussion

### 2.1. Designing the Experimental Set-Up

The combination of multiple spectroscopic techniques is a commonly used approach for gaining complementary information about a material's structure. The element selectivity of XAS spectra allows one to follow the evolution of an active metal species (here, platinum) by extracting its local atomic and electronic structure. UV–Vis spectroscopy can be used as both a qualitative and quantitative tool that is sensitive to the electronic structure of the compounds present in the sample. Mass spectrometry is often used in operando experiments to track the catalytic performance of the reaction under study.

To perform the combined laboratory XAS/UV–Vis/MS characterization, an operando photocatalytic cell was designed and produced, as shown in Figure 1a,b. The cell was equipped with two identical windows (4) made of scotch tape, which were transparent in both X-ray and UV ranges. The distance between the two windows, i.e., the thickness of the solution, was 4.6 mm. One of the windows (referred as the front window) was used

for carrying the $TiO_2$ support and the $Pt/TiO_2$ catalyst, while the second one (the back window) was used for UV irradiation. Initially, the ink with $TiO_2$ support was applied to the front window, and, after its complete drying, the cell was closed and filled with a solution of $K_2PtCl_6$ in a 1:3 mixture of formic acid and water. Two plastic pipes (5) were used to flush the cell with Ar continuously to avoid the presence of atmospheric oxygen. The cell was mounted inside a Rigaku R-XAS spectrometer, as shown in Figure 1c, adopting its fluorescence geometry. The incoming X-ray beam arrived from the left at ca. 45° with respect to the front window, while the fluorescence detector was located above the sample at ca. 90° with respect to the incoming beam. The 370 nm UV source with a remotely controlled digital power supply was applied from the back window. Pt $L_3$-edge XAS spectra were measured in fluorescence mode.

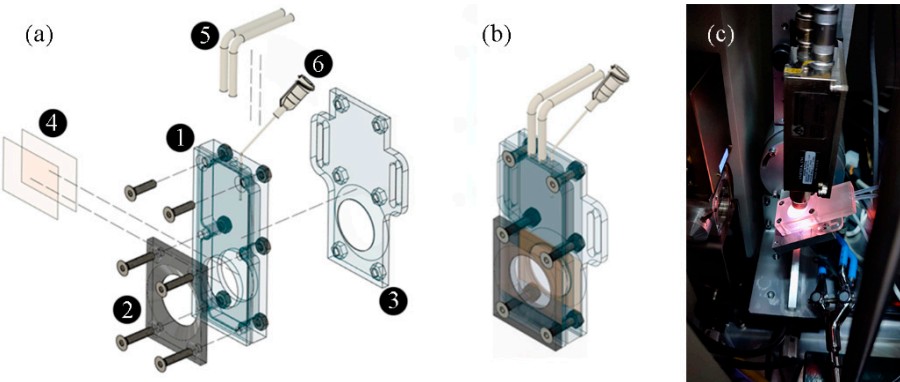

**Figure 1.** Schematic of the photocatalytic cell (**a**) and (**b**) with its main parts: 1—main body, 2—front window cap, 3—back window cap with fixture for Rigaku sample holder, 4—front and back windows, 5—pipes for gas inlet and outlet, 6—needle for liquid inlet/outlet. (**c**) Photo of the cell installed in the Rigaku R-XAS spectrometer during the photocatalytic experiment.

### 2.2. Insights on the Pt Species Photodeposited on $TiO_2$

The first challenge was to track the evolution of the oxidation state of Pt throughout the course of photodeposition. At the start of the experiment, Pt was in its cationic form in a solution. However, the concentration of Pt in the solution adopted during the photodeposition experiment was too low to collect high-quality XAS data, which is further complicated by the attenuation length of Pt $L_3$-edge energy photons in water of only about 3 mm. For this reason, a reference $K_2PtCl_6$ salt was measured ex situ (deposited on the window in the same setup but without the solution and $TiO_2$) resulting in a characteristic $Pt^{4+}$ XANES spectrum as shown via the black line in Figure 2a.

A second challenge derives from the use of a laboratory source for collecting XAS spectra. A significant amount of time is required to collect a single spectrum compared with that of synchrotron sources, which is not compatible with the time resolution needed to follow the Pt photodeposition process. Therefore, on the basis of the $K_2PtCl_6$ reference spectrum, three energy points were chosen (A, B, and C in Figure 2a): before the edge (11540 eV), at the white line (11560 eV), and after the white line (11580 eV). During the in situ Pt photodeposition from the solution onto the $TiO_2$ support under UV irradiation, the fluorescence signal was continuously collected in only these three points. After 2 h of irradiation, the UV source was switched off, and a complete XANES spectrum was collected in situ (Figure 2b, violet curve). This spectrum is characterized by a significantly reduced intensity of the white line at 11560 eV, with respect to that of $K_2PtCl_6$, and is characteristic of the Pt(0) state.

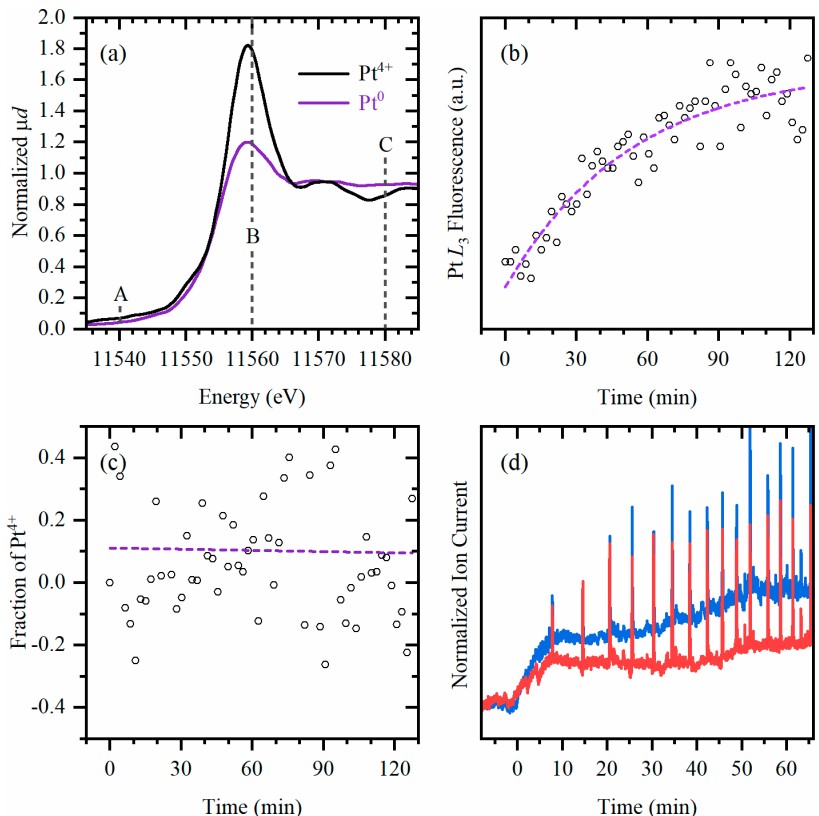

**Figure 2. (a)** Pt $L_3$-edge XANES of solid $K_2PtCl_6$ (black) and $Pt/TiO_2$ at the end of in situ photodeposition experiment (purple). The dashed vertical lines highlight the energy points in which the time-resolved fluorescence was collected. **(b)** Evolution of the absolute intensity of the fluorescence Pt signal in points B and C over time during the in situ photodeposition experiment, fitted by an exponential function (purple dashed line). **(c)** Estimation of $Pt^{4+}$ fraction based on Equation (1), fitted by a linear function (purple dashed line). **(d)** MS signal of $m/Z = 2$ (blue) and $m/Z = 44$ (red) divided by the signal of $m/Z = 40$ and normalized by area.

The relative fraction, $\gamma$, of the $Pt^{4+}$ species that contributes to the fluorescence counts can be estimated based on the relative intensity of the white line (point B), following Equation (1):

$$\gamma = \left( \frac{I_B - I_A}{I_C - I_A} - \frac{I_B^0 - I_A^0}{I_C^0 - I_A^0} \right) \Big/ \left( \frac{I_B^{4+} - I_A^{4+}}{I_C^{4+} - I_A^{4+}} - \frac{I_B^0 - I_A^0}{I_C^0 - I_A^0} \right) \tag{1}$$

where $I_A$, $I_B$, and $I_C$ are the intensities of the fluorescence signal at 11540, 11560, and 11580 eV, respectively, measured for the in situ sample (I), metallic Pt reference ($I^0$), and $K_2PtCl_6$ reference ($I^{4+}$). As can be seen from Figure 2c, the $Pt^{4+}$ fraction on $TiO_2$ was negligible at every stage of the experiment. Within the standard deviation, it was close to zero and did not show any trend. It should be also noted that the attenuation length of the Pt $L_3$-edge energy for $TiO_2$ was ca. 50 μm, while the estimated thickness of the $TiO_2$ layer was below 5 μm. Thus, the collected XAS signal is representative of the whole thickness of the $TiO_2$ layer, while is not informative about the Pt precursor present in the bulk of the solution. Based on the above, it can be concluded that only $Pt^0$ species were found on the $TiO_2$ layer during the whole photodeposition experiment.

Simultaneously with XAS data acquisition, $H_2$ evolution was monitored by online MS and is reported in Figure 2d. The signal of hydrogen was observed at the beginning and reached a value close to its maximum after only the first 10 min of experiment, even though the Pt photodeposition was still in progress and required a much longer time to be completed. This is evident in Figure 2b, which reports the evolution of the integral fluorescence signal over time proportional to the amount of platinum photodeposited onto

the $TiO_2$. A less significant increase in $H_2$ production activity after the first 10 min may be explained by the fact that the parallel increase of the fluorescence signal, proportional to the amount of Pt in the $TiO_2$, is associated with the growth of existing particles and not their quantity. Additional ex situ experiments were performed to better assess the photocatalytic performance of the system (Figure S1 from Supplementary Materials). The $H_2$ evolution rate was about 3.5 mmol $g_{cat}^{-1}$ $h^{-1}$, with a similar rate for $CO_2$ production, as would be expected considering the stochiometric ratio in the formic acid photoreforming reaction. The observed rate is comparable to other Pt-$TiO_2$ systems reported in literature [34,35].

### 2.3. Insights on the Pt Species in Solution

Every 6 min during the photodeposition experiment, small (ca. 1 mL) aliquots of the solution were extracted for UV–Vis measurements and were immediately returned to the cell after a spectrum was collected. The spectrum of the initial solution (black) (Figure 3), measured against distilled water as a reference, had two bands at 260 nm and in the 200–250 nm range, corresponding to the charge transfer transition involving the $Pt^{4+}$ species and absorption by formic acid, respectively. Notably, the latter band had already disappeared from the collected spectrum after 6 min of UV irradiation, and no significant changes were observed over the following 90 min. Formic acid absorption also decreased over time. At the same time, the fluorescence signal of the $Pt^0$ on $TiO_2$ grew over a significantly larger time scale (Figure 2b), indicating that only a minor part of the reduced $Pt^{4+}$ species are deposited on the $TiO_2$ support in the first 6 minutes of the reaction.

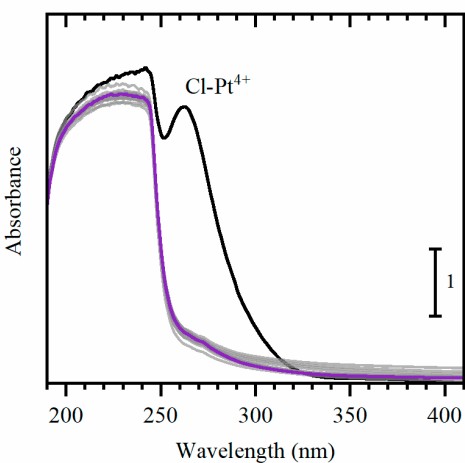

**Figure 3.** Evolution of UV–Vis spectra of the $K_2PtCl_6$ solution during photodeposition of Pt on $TiO_2$. The spectra before and after 90 min UV irradiation are shown in black and violet, respectively; intermediate data are shown in light grey.

An additional experiment was performed to investigate the fate of the Pt species in the solution without the addition of $TiO_2$ for Supplementary Materials (Figure S2). The experiment was carried out with a Pt concentration that was lower than the previous experiment by a factor of 5. The results (Figure S2a) confirmed a rapid decrease in the $Pt^{4+}$ signal after only 30 s of exposure. Additionally, it was shown that the presence of titanium dioxide was not necessary in this step, meaning that the reduction of the $K_2PtCl_6$ precursor is exclusively a result of UV irradiation (Figure S2b).

This finding also explains why the reduction of the $Pt^{4+}$ precursor in the solution, as monitored by UV–Vis spectroscopy, occurred before its deposition on $TiO_2$, as observed by XAS. However, we did not observe the formation of Pt nanoparticles in the solution, which would have provided a broad spectral feature in the UV–Vis spectra, arising from light scattered by nanoparticles.

For this reason, we performed an experiment where the same amount of $K_2PtCl_6$ precursor was dissolved in a mixture of deionized water and formic acid (3:1) and irradiated

by UV light. As can be seen in Figure 4a, the $Pt^{4+}$ signal also disappeared after 1 h of irradiation (from black to purple). The UV–Vis data were measured against the solution of formic acid and water as a reference to exclude formic acid absorption from the resulting spectrum. After UV irradiation, the solution became dark and provided an increased background to the UV–Vis spectra, which can be attributed to the scattering by Pt nanoparticles in the solution [36]. To support this fact, the solution was dried, and the remaining fraction was probed by X-ray diffraction (XRD). Broad peaks of *fcc* platinum with a cell parameter of 3.9194 (4) Å co-exist with the reflections from KCl salt with a cell parameter of 6.3011 (5) Å (Figure 4b). The average crystallite size of the platinum particles, determined according to the *LX* parameter of the Lorentzian-broadening dependence on the scattering angle, was 3.2 nm. As a side note, although the reduction of $Pt^{4+}$ was observed in the absence of $TiO_2$, no signal of hydrogen production was observed for such a sample. However, a signal of $CO_2$ was registered at the beginning of irradiation as soon as the UV light was on, evidencing its role in the reduction of platinum.

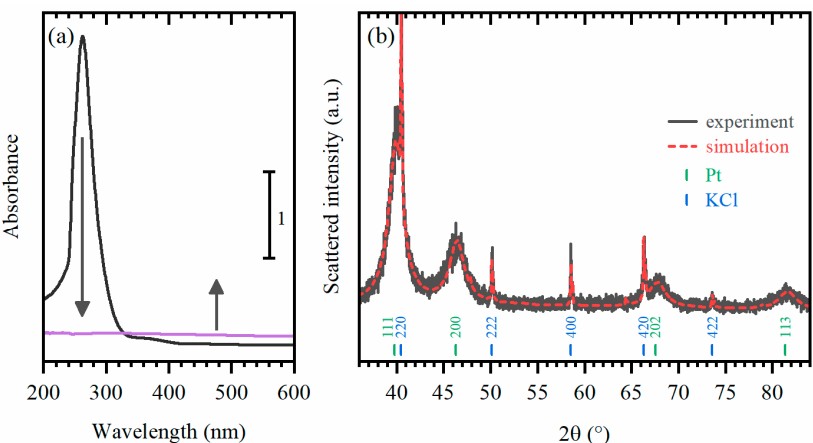

**Figure 4.** (**a**) UV–Vis spectra of $K_2PtCl_6$ in the solution of water and formic acid (3:1) without $TiO_2$ before (black) and after 1 h of UV irradiation (purple). (**b**) Experimental (solid black) and simulated (dashed pattern) intensities of the dried sample after UV irradiation.

## 3. Materials and Methods

### 3.1. Photocatalytic Cell

3D, solid models of the cell components (Figure 1a) were designed using Fusion 360 (Autodesk, San Rafael, CA, USA). The solid models were then converted into STL files for 3D printing. The cell for operando spectroscopic investigation was 3D printed using the digital-light-processing 3D printer Asiga MAX UV (Asiga, Sydney, Australia) with a wavelength of 385 nm and a light intensity of 7.25 mW/cm². The first layer was set at 25 μm and was exposed for 20 s to avoid the delamination of the print from the platform. The thickness of the layers was set to 25 μm and each layer was exposed for 1.1 s. To avoid the delamination of the layers during the process, the *z*-compensation was set to 300 μm. For a better processability of the resin during the print, the printing temperature was set to 47 °C. Immediately after printing, the cell was sonicated in IPA for one minute at 80 kHz and then mounted in the holder for manual flushing with IPA. After the flushing, the cell components were sonicated once more and blow dried using nitrogen gas. Lastly, the cell was post-cured for 2 min using a UV-radiation lamp (Flash DR-301C, Asiga, Sydney, Australia).

### 3.2. Experimental Procedure

In a typical procedure, a solution of formic acid and deionized water (1:3) was prepared and stirred for 10 minutes at room temperature. A total of 2 mg of $K_2PtCl_6$ (Sigma–Aldrich, St. Louis, MO, USA) was then added to 13.3 mL of the prepared solution and stirred for 30 minutes. The $TiO_2$ ink was prepared by mixing ca. 7 mg of $TiO_2$ (634662 Sigma–Aldrich,

St. Louis, MO, USA, Titanium (IV) oxide, with a mixture of rutile and anatase, nanopowder, a <100 nm particle size (BET), and a 99.5% trace metals basis) with 100 μl of deionized water. The ink was deposited on the window of the photocatalytic cell and left drying for 15 min. The solution of $K_2PtCl_6$ (ca. 10 mL) was then added, and the closed cell was bubbled with Ar (20 mL/min for 15 min). Then, the gas inlet was adjusted to flush only the headspace of the cell in order not to create bubbles in the liquid during the measurements. An LED–UV light source (370 nm, 3.9 W) was applied from the window opposite from the $TiO_2$ layer.

### 3.3. XAS Data Collection

The Pt $L_3$-edge X-ray absorption near-edge structure (XANES) spectra were measured in the operando regime using an R-XAS Looper (Rigaku, Japan) laboratory X-ray absorption spectrometer at the Smart Materials Research Institute of the Southern Federal University. The measurements were performed in fluorescence yield geometry. The energy was selected by a Si (620) Johansson bent monochromator, providing an energy resolution of $\Delta E$ = 1.5 eV for Pt $L_3$-edge energy (11564 eV). The incident beam intensity was measured by an Ar-filled (300 mbar) ionization chamber, and the fluorescence signal was measured by a silicon drift detector. For the time-resolved experiment, the data were collected at 3 points: before the edge, at the maximum of the white line of the $K_2PtCl_6$ reference, and after the white line (see Figure 4). The data were processed using the Athena software [37].

### 3.4. UV–Vis Data Collection

UV–Vis spectra were measured using a Shimadzu UV-2600 spectrophotometer (Shimadzu Co., Kyoto, Japan). A photocatalytic cell was equipped with a needle outlet, through which aliquots of the solution (ca. 1 mL) were taken to collect the UV–Vis spectrum (ca. 3 min) and then returned to the cell. The sample and the reference (distilled water) were contained in two quartz cuvettes of 5 mm thickness. First, the UV–Vis spectrum was collected from a solution purged for 15 minutes with Ar. Then, starting from the moment of UV light irradiation, a probe was taken every 6 minutes and, upon completion of spectrum collection, was returned back to the cell. For all data, the spectral acquisition was performed with a 0.5 nm step in the 600–185 nm range.

### 3.5. Mass Spectrometry Data Collection

MS data was collected using the online quadrupole mass spectrometer TekhMas 7–100 (AtomTyazhMash, St. Petersburg, Russia). The device was attached via T-connection to the gas outlet of the photocatalytic cell to monitor the signal of the produced $H_2$ ($m/Z$ = 2). The signals of Ar, water, oxygen, nitrogen, and formic acid were also tracked. For better statistics, the $H_2$ signal was averaged over 90 s prior to the time when the aliquots were extracted for UV–Vis collection.

### 3.6. X-ray Diffraction Measurements

XRD patterns were collected using the Bruker D2 PHASER instrument (Bruker, Billerica, MA, USA) with a Cu $K_\alpha$ source. A sample for XRD measurement was prepared by dissolving 20 mg of $K_2PtCl_6$ into a 133 mL solution of formic acid and deionized water (1:3). The solution was irradiated by UV light for 1 h and then dried overnight. The remaining solid fraction was put on a low-background sample holder. The XRD data were collected using Bragg–Brentano geometry in the 2θ range, from 20–90°, with a step of 0.01°. The Pawley fitting was performed in January 2006 [38].

## 4. Conclusions

Although being extremely informative regarding the structure of active sites, the operando XAS technique is much less used in photocatalytic studies compared with the conventional heterogenous catalysis. This is generally due to the technical complexity of the sample environment necessary to perform photocatalytic experiments compatible with

XAS data collection. In this regard, our study represents an example of how photocatalysts can be studied in situ, even at the laboratory scale, without the utilization of synchrotron light. In combination with optical spectroscopy and mass spectrometry data, several new conclusions were made concerning the photodeposition of Pt on $TiO_2$ support and the successive HER reaction over the formed $Pt/TiO_2$ catalyst.

The amount of photodeposited Pt on $TiO_2$ identified as Pt(0) grew progressively over ca. 1 h, while the signal of the produced $H_2$ was detected and saturated mainly in the first 10–20 min. At the same time, the characteristic $Pt^{4+}$ peak of the precursor in the solution for the UV–Vis data disappeared immediately as soon as the UV light was switched on in the presence of formic acid acting as a hole scavenger. The reduction of $Pt^{4+}$ in the solution to Pt(0) nanoparticles by UV irradiation was also confirmed in absence of $TiO_2$.

These results provide useful insight into the formation and evolution of $Pt/TiO_2$ photocatalysts and suggest an effective multi-technique operando approach for both laboratory and synchrotron-based studies of water splitting and HER photocatalysts.

**Supplementary Materials:** The following supporting information can be downloaded at: https://www.mdpi.com/article/10.3390/catal13020414/s1; Figure S1—formation of $H_2$ and $CO_2$ during PHER, analyzed by GC.; Figure S2—evolution of UV–Vis spectra; Figure S3—normalized MS signal of $m/Z = 44$; Table S1—Pt content from XRF elemental analysis.

**Author Contributions:** Conceptualization, A.L.B., E.G. and L.M.; methodology, A.L.B.; investigation, E.G.K., P.N.N., S.V.C., V.V.S., A.Y.P., A.N.B., A.A.S. and F.P.; data curation, E.G.K. and P.N.N.; writing—original draft preparation, E.G.K. and A.L.B.; supervision, A.L.B. and L.M.; funding acquisition, A.V.S. and S.B. All authors have read and agreed to the published version of the manuscript.

**Funding:** This work was financially supported by the Ministry of Science and Higher Education of the Russian Federation (Agreement № 075-15-2021-1389 from 13 October 2021). L.M, F.P., E.G. and S.B. thank the Italian Ministry of Foreign Affairs for financial support through the project "Nanocatalysts for photostimulated green hydrogen production: molecular design and advanced characterization assisted by machine learning approaches".

**Data Availability Statement:** The data presented in this study are available on request from the corresponding author. The data are not publicly available as they are the property of the Southern Federal University.

**Acknowledgments:** We acknowledge the Ministry of Science and Higher Education of the Russian Federation for financial support (Agreement № 075-15-2021-1389 from 13 October 2021).

**Conflicts of Interest:** The authors declare no conflict of interest. The funders had no role in the design of the study; in the collection, analyses, or interpretation of data; in the writing of the manuscript; or in the decision to publish the results.

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
