# Peer review of "Operando Laboratory X-ray Absorption Spectroscopy and UV–Vis Study of Pt/TiO2 Photocatalysts during Photodeposition and Hydrogen Evolution Reactions"

_catalysts, doi:10.3390/catal13020414_

Round 1

Reviewer 1 Report

The authors report the use of multi-technique approach to study catalyst formation and photocatalytic H2 production over Pt supported on TiO2 combining laboratory analysis including: The X-ray absorption spectroscopy (XAS), UV-Vis spectroscopy and mass spectrometry (MS). The authors present a home-made 3D printed cell that was used to perform an operando characterization. This is an interesting study combining multiple spectroscopic techniques in one experimental setup provides that enables the possibility to look at catalytic systems from different perspectives by getting complementary information. They have clearly provided sufficient background for the better understanding of their investigation. The methodology is designed appropriately and the results are clearly presented. However I will suggest to the authors to split Figure 2 into two distinct figures. 

I recommend the acceptance of this manuscript in the present form in catalyst. 

Author Response

We thank Reviewer for his/her positive report. According to the made suggestion we have updated figure 2 in the revised version of the manuscript. It still remains as a single Figure but is now made in two rows. Also, the MS data of H2 signal was complemented with CO2 one.

Reviewer 2 Report

The research entitled “Operando laboratory X-ray absorption spectroscopy and UV-Vis study of Pt/TiO2 photocatalyst during photodeposition and hydrogen evolution reaction” reveals the application of operando spectroscopy in the fields of in situ reactions. I recommend the research be published in the journal after a major revision.

1. Introduction can be improved some extent.

2. The Pt references on Operando studies could be added in the introduction.

3. FESEM and TEM images should be provided.

4. Explain the Operando studies with Raman and XPS results in addition to the XRD.

5. Provide about the PHER efficiency of the catalysts

6. Compare the results with the literature reports.

Author Response

We thank Reviewer 2 for his report. In the attached file, we copy the Reviewer’s comments (in black) followed by our replies (in red). We also attach to this submission the manuscript file, where all the changes are highlighted in Trach Changes mode and Supporting Information file, which was not present in the initial submission.

Reviewer 3 Report

The present work is a study on the formation of platinum on titanium dioxide and the photocatalytic production of hydrogen in the operando mode. Despite the fact that the photocatalytic production of hydrogen on platinized titanium dioxide has been studied for years, the use of the operando mode allows you to look "deep" into the process and understand how, for example, platinum is reduced. Based on this, the article is of great interest. There is only one small remark: what is the standard deviation of the values indicated in Figure 2c. Does it make sense to build dependencies with such a large spread of points?

Author Response

We thank Reviewer for the positive report.

Indeed, the spread of points in Figure 2c is large, with the standard deviation of 0.2. Even in the pristine manuscript, we did not tried to speculate about any dependencies, which are within the standard deviation, and only claimed that after the photodeposition, platinum is in Pt(0) oxidation state without significant evolution during the experimental procedure. We tried to state this more clearly in the revised version.

Reviewer 4 Report

The manuscript entitled “Operando laboratory X-ray absorption spectroscopy and UV-Vis study of

Pt/TiO2 photocatalyst during photodeposition and hydrogen evolution reaction” describes the investigation of Pt deposition on TiO2 and its hydrogen generation. Although the idea is interesting, sufficient data is lacking and the manuscript sound repetitive. There are many things need to be clarified and therefore, looking at this current manuscript, rejection is recommended.

Some of the comments are as follow:

·        Section 2.1 describe the experimental set-up and should not be in results and discussion section.

·        In operando studies, description of the mechanism is important and none is included.

·        In the statement “The signal of hydrogen is observed since the beginning and reaches already a value close to its maximum after the first 10 min of experiment, even though the Pt photodeposition is still in progress and requires much longer times to be completed. This is evident in Figure 2c, which reports the evolution of the integral flu-130 orescence signal over time, which is proportional to the amount of platinum photodeposited onto TiO2”.
It is difficult to comprehend. Reference is Figure 2 C is not proportional as the graph does not show a trend. The description of H2 generation reaching a max after 10 min is redundant as it is also mentioned again in later part, “The H2 signal monitored by MS grows significantly in the ca. 10 min of the photodeposition (Figure 2d), and then a further moderate growth was observed.”

·        The authors mentioned that “parallel increase of the fluorescence signal, proportional to the amount of Pt in TiO2, is associated with the growth of the existing particles, and not of their quantity”. Such claim should be proven with SEM or TEM.

·        “..indicating that only a minor part of the reduced Pt4+ species are deposited on the TiO2 support in the first 6 minutes of reaction.” Such claim should be justified with more characterization like SEM and TEM.

·        Figure 3 with clear indication of the transition should be included. It seems the interval UV-Vis results also exceeded the final absorbance results.

·        Proof on CO2 detection should be included.

·        What is the error bar in Fig.3 and 4(a). The simulation in Fig,4(b) is not clearly explained

Author Response

We thank Reviewer 4 for his report. In the attached file, we copy the Reviewer’s comments (in black) followed by our replies (in red). We also attach the manuscript file, where all the changes are highlighted in Trach Changes mode and Supporting Information file, which was not present in the initial submission.

Round 2

Reviewer 2 Report

I appreciate the authors kind reply. The manuscript can be accepted now.

Reviewer 4 Report

As the reviewer mentioned in the first report, the idea of operando study of this work is interesting and lacking sufficient data. However, the authors have made significant improvement to the manuscript as well as responded to the comments. Additional clarification of the interval of UV-Vis with better time resolution and CO2 detection results have also been added. 

Therefore, the reviewer now feels the emphasis of this manuscript on the feasibility of operando investigation of Pt/TiO2 is sound and acceptable. The effort and persistence is appreciated.